# Effects of a 12-Month Treatment with Glucagon-like Peptide-1 Receptor Agonists, Sodium-Glucose Cotransporter-2 Inhibitors, and Their Combination on Oxidant and Antioxidant Biomarkers in Patients with Type 2 Diabetes

**DOI:** 10.3390/antiox10091379

**Published:** 2021-08-28

**Authors:** Vaia Lambadiari, John Thymis, Dimitris Kouretas, Zoi Skaperda, Fotios Tekos, Foteini Kousathana, Aikaterini Kountouri, Konstantinos Balampanis, John Parissis, Ioanna Andreadou, Maria Tsoumani, Christina Chania, Konstantinos Katogiannis, George Dimitriadis, Aristotelis Bamias, Ignatios Ikonomidis

**Affiliations:** 12nd Department of Internal Medicine, Research Unit and Diabetes Centre, Attikon Hospital, Medical School, National and Kapodistrian University of Athens, 12462 Athens, Greece; vlambad@otenet.gr (V.L.); f.kousathana@hotmail.com (F.K.); KaterinaK90@hotmail.com (A.K.); kostasbalabanis@gmail.com (K.B.); gdimitr@med.uoa.gr (G.D.); abamias@med.uoa.gr (A.B.); 22nd Department of Cardiology, Attikon Hospital, Medical School, National and Kapodistrian University of Athens, 12462 Athens, Greece; johnythg@gmail.com (J.T.); jparissis@yahoo.com (J.P.); kenndj89@gmail.com (K.K.); 3Department of Biochemistry and Biotechnology, University of Thessaly, 41500 Larissa, Greece; dkouret@uth.gr (D.K.); zoiskap94@gmail.com (Z.S.); fotis.tek@gmail.com (F.T.); 4Laboratory of Pharmacology, Faculty of Pharmacy, National and Kapodistrian University of Athens, 15771 Athens, Greece; jandread@pharm.uoa.gr (I.A.); marietsoumani@gmail.com (M.T.); cgchania@gmail.com (C.C.)

**Keywords:** glucagon-like peptide-1 receptor agonists, sodium-glucose cotransporter-2 inhibitors, Malondialdehyde, Thiobarbituric Acid Reactive Substances, 2,2¢-azino-bis-(3-ethylbenzthiazoline-6-sulphonic acid) radical

## Abstract

Imbalance between oxidative stress burden and antioxidant capacity is implicated in the course of atherosclerosis among type 2 diabetic patients. We addressed the effects of insulin, glucagon-like peptide-1 receptor agonists (GLP1-RA), sodium-glucose cotransporter-2 inhibitors (SGLT-2i), and their combination on levels of oxidant and antioxidant biomarkers. We recruited a total of 160 type 2 diabetics, who received insulin (*n* = 40), liraglutide (*n* = 40), empagliflozin (*n* = 40), or their combination (GLP-1RA+SGLT-2i) (*n* = 40). We measured at baseline, at 4 and at 12 months of treatment: (a) Thiobarbituric Acid Reactive Substances (TBARS), (b) Malondialdehyde (MDA), (c) Reducing Power (RP), (d) 2,2¢-azino-bis-(3-ethylbenzthiazoline-6-sulphonic acid) radical (ABTS) and (e) Total Antioxidant Capacity TAC). Dual treatment resulted in significant improvement of TBARS, MDA, and ABTS at four months compared with the other groups (*p* < 0.05 for all comparisons). At twelve months, all participants improved TBARS, MDA, and ABTS (*p* < 0.05). At 12 months, GLP1-RA and GLP-1RA+SGLT2-i provided a greater reduction of TBARS (−8.76% and −9.83%) compared with insulin or SGLT2i (−0.5% and 3.22%), (*p* < 0.05). GLP1-RA and GLP-1RA+SGLT-2i showed a greater reduction of MDA (−30.15% and −31.44%) compared with insulin or SGLT2i (4.72% and −3.74%), (*p* < 0.05). SGLT2i and GLP-1RA+SGLT2-i showed increase of ABTS (12.87% and 14.13%) compared with insulin or GLP1-RA (2.44% and −3.44%), (*p* < 0.05). Only combined treatment resulted in increase of TAC compared with the other groups after 12 months of treatment (*p* < 0.05).12-month treatment with GLP1-RA and SGLT2i resulted in reduction of biomarkers responsible for oxidative modifications and increase of antioxidant biomarker, respectively. The combination treatment was superior and additive to each separate agent and also the beneficial effects appeared earlier.

## 1. Introduction

The incidence of Type 2 Diabetes Mellitus is increasing rapidly worldwide resulting in increased atherosclerotic cardiovascular complications and consequently excess cardiovascular morbidity and mortality in diabetics compared to the general population [1,2,3,4,5]. Oxidative stress confers remarkably to the detrimental cardiovascular effects of DM by enhancing atherosclerotic processes [6,7]. Six distinct metabolic pathways have been found to contribute to reactive oxygen (ROS) production in diabetes because of sustained hyperglycemia namely: (1) enolization and α-ketoaldehyde formation, (2) PKC activation, (3) dicarbonyl formation and glycation, (4) sorbitol metabolism, and (5) hexosamine metabolism increasing the oxidative burden and tissue damage including b-pancreatic cells [8]. Two novel antidiabetic classes of agents, SGLT2i and GLP1-RA have been approved for diabetic patients who pose high cardiovascular risk, and hence they are advocated even as a first line treatment in patients at risk by the running guidelines [9]. These agents exert their favorable cardiovascular actions in different ways; GLP1-RA are mainly considered to have anti-atherogenic and anti-inflammatory properties, while SGLT-2i act primarily on vascular hemodynamics, but they are both proposed to reduce oxidative stress burden and enhance antioxidant capacity [10,11,12].

However, the impact of the above-mentioned agents on Thiobarbituric Acid Reactive Substances (TBARS), Malondialdehyde (MDA), Reducing Power (RP), 2,2¢-azino-bis-(3-ethylbenzthiazoline-6-sulphonic acid) radical (ABTS) and Total Antioxidant Capacity (TAC) has not been fully clarified yet. TBARS and MDA are biomarkers that depict the degree of lipid peroxidation and are used for assessment of oxidative stress. They are also associated with increased incidence of cardiovascular disease [13,14,15]. Additionally, TBARS and MDA are found affected in patients with T2DM and non-alcoholic fatty liver disease and are also linked with glucose levels and obesity [16,17,18,19]. Lipid peroxidation products are relatively unstable and are the result of a single specific oxidant in contrast to protein carbonyls that represent an irreversible form of protein modification, are formed early during oxidative stress conditions, are relatively stable and are not a result of one specific oxidant and thus they are considered an accurate marker of overall protein oxidation [20].

TAC, a reliable biomarker of antioxidant defense, influenced by uric acid levels has been found reduced in obese and diabetic patients in many large studies, although some smaller have not described that distinction [19,21,22,23,24,25,26,27,28]. Uric acid appears to be a strong independent predictor for TAC causing a great increase in TAC if its concentration is elevated in plasma. Vitamin C is the second strongest predictor of TAC followed by vitamins E and A. Moreover, ABTS and RP constitute commonly used antioxidant biomarkers, but there are gaps in bibliography regarding their role in DM [15,29,30]. Consequently, TAC, RP, and ABTS serve as biomarkers of serum antioxidant activity, whereas TBARS quantifies oxidative stress burden.

We have previously shown that a 12-month treatment with the combination resulted in a greater improvement on markers of arterial stiffness, endothelial glycocalyx thickness, ventricular-arterial coupling and myocardial work [31]. Nonetheless, it remains unclear to our knowledge whether combination of GLP-1RA and SGLT-2i has a synergistic and consequently more favorable effect on markers of oxidative stress burden and antioxidants respectively, rendering a beneficial equilibrium of oxidants and antioxidants.

In the present study of type 2 diabetics with high or very high cardiovascular risk, we speculated that oxidative stress burden -assessed by serum levels of TBARS and MDA- and antioxidant biomarkers -via estimating the serum levels of ABTS, TAC, RP- could improve after treatment with combination of GLP-1RA and SGLT-2i compared to the separate administration of each agent or treatment with insulin.

For this reason, we documented prospectively the changes of TBARS, ABTS, TAC, RP, before and after 4 and 12 months of treatment in four parallel groups of T2DM treated with insulin, GLP-1RA, SGLT-2i and the combination GLP-1RA and SGLT-2i as a second step treatment to metformin for one year correspondingly.

## 2. Materials and Methods

### 2.1. Study Design

We initially recruited a total of 180 individuals with T2DM from the Cardiometabolic Outpatient Clinic of Attikon University hospital who posed high or very high cardiovascular risk, as previously published [31]. Patients with high CV risk were considered those with T2DM and a calculated SCORE ≥ 5% and <10%, while very high CV risk those with T2DM and target organ damage or with a major risk factor such as smoking or poorly controlled hyperlipidemia or poorly controlled hypertension (a calculated SCORE ≥ 10%) [31,32]. The analysis was conducted by two observers, who were blinded for the data. Patients were randomly assigned to one of the following regimens: (1) basal insulin, with a dose range 10 IU to 50 IU and the dose was titrated according to the running guidelines of that period, (45 of the participants), (2) 1.8 mg of the GLP1-RA liraglutide subcutaneously once daily, which was escalated weekly (45 of the participants), (3) 25 mg of the SGLT2i empagliflozin orally once daily (45 of the participants), and (4) or the combination liraglutide + empagliflozin as a second step treatment to metformin [31,33,34,35]. Assignment to a treatment regimen by an attending physician (F.K) using a table of random numbers as reproduced from the online randomization software http://www.graphpad.com/quickcalcs/index.cfm (accessed on 3 November 2017). The participants were followed up strictly for 12 months, and reevaluation took place at the outpatient clinic at 4 and 12 after initial enrollment, respectively.

Exclusion criteria included active neoplasia, chronic inflammatory autoimmune disease, chronic kidney disease (estimated GFR less than 60 mL/min/1.73 m^2^ for a period of at least 90 days), severe liver disease, peripheral vascular disease and retinopathy. None of the female patients received hormone replacement treatment during recruitment or at follow-up. More specifically, all patients suffered from hyperlipidemia and were on statins. Hypertension was defined as brachial blood pressure > 140/90 mmHg or treatment with antihypertensive agents.

20 subjects were totally excluded from the final analysis. More precisely, 5 patients were excluded from the insulin group, out of which 3 owing to unwillingness to inject themselves and 2 could not be reached to complete the follow-up. Also, in GLP-1RA arm 4 individuals discontinued treatment due to GI disturbances and 1 subject lost at follow-up; in SGLT-2i group 4 participants stopped treatment because of frequent genitourinary tract infections and 1 subject lost at follow-up and in dual therapy group 2 subjects discontinued due to severe gastrointestinal symptoms, 2 patients owing to genitourinary tract infection and 1 participant lost at follow-up. As a result, 160 subjects were finally included in the analysis. Figure 1 depicts a comprehensive chart of study progress. The primary endpoints of our study were to document the changes of oxidative burden biomarkers-namely TBARS and MDA serum levels- and antioxidant biomarkers -namely ABTS, TAC, RP- after 12-month administration of the combination GLP1-RA+SGLT2i and to compare with the separate administration of each agent or treatment with insulin.

All subjects signed an informed consent. The protocol was aligned with the principles outlined in the Declaration of Helsinki and was approved by the hospital’s Ethics Committee.

Blood sampling took place at the Cardiometabolic outpatient clinic of Attikon Hospital.

Initially, a detailed medical history was recorded for each participant and then every individual underwent a physical, clinical, echocardiography examination by the medical personnel and afterwards blood was drawn. This process was repeated in every follow-up.

### 2.2. Laboratory Assays

#### 2.2.1. TBARS

Oxidative stress in the cellular environment results in the formation of highly reactive and unstable lipid hydroperoxides. Decomposition of the unstable peroxides derived from polyunsaturated fatty acids results in the formation of malondialdehyde (MDA), which can be quantified following its controlled reaction with thiobarbituric acid (TBA). TBARS assay was used for the determination of lipid peroxidation and a slightly modified assay of Keles et al. was used [36]. In particular, 100 μL of plasma was mixed with 500 μL of 35% TCA and 500 μL of tris-hydroxy-methyl-aminomethane hydrochloride (Tris–HCl) (200 mM, pH = 7.4) and incubated for 10 min at room temperature. 1 mL of 2 M sodium sulfate (Na2SO4) and 55 mM thiobarbituric acid (TBA) solution was added, and the samples were incubated at 95 °C for 45 min. The samples were cooled on ice for 5 min and were vortexed after adding 1 mL of 70% TCA. The samples were centrifuged at 15,000× *g* for 3 min at room temperature and the absorbance of the supernatant was monitored at 530 nm. A baseline shift in absorbance was taken into account by running a blank along with all samples during the measurement. TBARS are expressed as malondialdehyde (MDA) equivalents and the calculation of TBARS concentration was based on the molar extinction coefficient of MDA“ (156,000 (L/mol/cm)).

#### 2.2.2. MDA

MDA was estimated spectrophotometrically with a commercial kit (Oxford Biomedical Research, Rochester Hills, MI, USA) of colorimetric assay for lipid peroxidation (measurements range, 1–20 nmol/L) with a previously described methodology [37].

#### 2.2.3. ABTS

This method is used to assess the antioxidant capacity, based on the interaction between molecules with stable radical ABTS • +. The radical of ABTS • + is produced by the mixing 2 mM of 2,2-Azino-bis- (3-ethyl-benzthiazoline-sulphonic acid) (ABTS) with 30 μM H_2_O_2_ and 6 μM horseradish peroxidase (HRP) enzyme in 50 mM PBS (pH = 7.5) [38]. To assess the antioxidant activity of a substance must first be preceded by the formation of the radical and then followed by incubation with the test compound. Immediately, following the addition of the HRP enzyme, the contents were vigorously mixed, incubated at room temperature in the dark and the reaction was monitored at 730 nm until stable absorbance was obtained. Subsequently, 10 μL of plasma were added in the reaction mixture and the decrease in absorbance at 730 nm was measured. The radical of ABTS • + is a substance that has green color and absorbs at 730 nm. When adding to the solution a substance with antioxidant activity, then the radical of ABTS • + is reduced with the addition of a hydrogen atom, resulting in decrease of the optical absorbance at 730 nm [38].

#### 2.2.4. TAC

The term Total Antioxidant Capacity (TAC) is referred to the capability of the plasma components to scavenge reactive species. It is an indicator of the overall antioxidant capacity of plasma [21,22]. Every plasma component has an antioxidant activity by itself. However, each one contributes differently in the total antioxidant capacity of the plasma. It is a measure of the antioxidant status of the body in general. Determination of TAC was based on the method of Janaszewska and Bartosz. Briefly, 20 µL of plasma was added to 480 µL of 10 mM sodium potassium phosphate buffer (pH 7.4) and 500 µL of 0.1 mM 2,2-diphenyl-1-picrylhydrazyl (DPPH) free radical and samples were incubated in the dark for 60 min at room temperature. Samples were centrifuged for 3 min at 20,000× *g* and the absorbance was monitored at 520 nm. TAC is presented as mmol of DPPH reduced to 2,2-diphenyl-1-picrylhydrazine (DPPH:H) by the antioxidants of plasma.

#### 2.2.5. RP

Reducing power may serve as a significant reflection of the antioxidant activity. Compounds with reducing power indicate that they are electron donors and can reduce the oxidized intermediates of lipid peroxidation processes, so that they can act as primary and secondary antioxidants. The reducing power assay was based on Yen and Duh protocol, with slight modifications [29]. An aliquot (10 μL) plasma sample was dissolved in 240 μL phosphate buffer (0.2 M, pH = 6.6) and 250 μL of 1% potassium ferricyanide was added and the mixed was incubated at 50 °C for 20 min. The samples were cooled on ice for 5 min. Then, 250 μL of 10% TCA was added and the samples were centrifuged at 3000 rpm for 10 min in room temperature. Subsequently, 250 μL of dH_2_O and 50 μL of 0.1% ferric chloride were added to the supernatant and the samples were incubated in the dark for 10 min at room temperature. The absorbance was monitored at 700 nm. A sample without plasma and 1% potassium ferricyanide was used as blank and a sample only without plasma was used as control.

### 2.3. Statistical Analysis

Statistical analysis was conducted by SPSS version 26 (IBM SPSS Statistics, Inc., Chicago, IL, USA) and Stata version 16 (StataCorp LP, College Station, TX, USA). All continuous variables were presented as mean ± SD if they were distributed normally, as indicated by the Kolmogorov-Smirnov and Shapiro-Wilk normality tests. In case of non-normal distribution, transformation into ranks followed. Nominal variables were presented as percentages. Correlation analysis was performed via Spearman or Pearson correlation tests, based upon data distribution. Categorical variables were analyzed performing either Chi-square tests or Fisher’s exact tests, as appropriate. Analyses were intention to treat. ANOVA for repeated measurements was used for (1) measurements of the biomarkers at the beginning, at 4 and at 12 months of treatment (which was considered as a within-subject factor and (2) also the effects of different treatments, as a between-subject factor (the 4 treatment groups were the insulin group, the GLP-1RA group, the SGLT-2i, and finally the dual GLP-1RA+SGLT-2i group). Baseline BMI, level of blood lipids, glucose or HbA1 and their respective changes at 4 or 12 months were included as covariates in the model. Post-hoc comparisons were conducted with Bonferroni’s correction. The F and the respective *p* values between the different time periods of measurements were calculated. Furthermore, the F and *p* values between time of measurement of the biomarkers and the examined treatments were estimated. The F and the corresponding *p* values of the comparison between treatments were calculated also. The Greenhouse-Geisser, Huynh-Feldt or Lower-Bound corrections were used when the sphericity assumption, as evaluated by Mauchly’s test, was not met. Post-hoc comparisons were conducted with Bonferroni’s correction. The percent changes of the examined variables after intervention were also compared by ANOVA. All statistical tests were two-tailed and a *p*-value < 0.05 was considered statistically significant.

## 3. Results

Table 1 summarizes the baseline characteristics of our study population. Briefly, the mean age of the study population was 58 ± 10 years and 28% (45/160) of them were females. Moreover, approximately 1/3 of the participants had overt coronary artery disease and 61% suffered from hypertension. Of note, 109/160 (68%) were already receiving metformin at inclusion. All comparisons concerning the above-mentioned characteristics were not statistically significant between groups (*p* = NS).

### 3.1. Glycemic Control after Intervention

Baseline HbA1C and fating glucose was similar in all participants (*p* = NS). During follow-up all participants achieved significant reduction of HbA1C (F = 13.86, *p* < 0.001) and fasting glucose (F = 5.1, *p* = 0.013) and no significant distinctions between treatment groups were recorded (Table 2). BMI was decreased in the overall population after 4 and 12 months (*p* < 0.001; Table 2). There was a significant interaction between the type of treatment and the change of BMI posttreatment (F = 5.939, *P* for interaction = 0.002; Table 2). Patients treated with combination of GLP-1RA+SGLT-2i presented a greater percentage reduction of BMI compared with GLP-1RA (*p* = 0.042) or SGLT-2i (*p* = 0.009; Table 2) Total cholesterol, low-density lipoprotein cholesterol, and triglycerides blood levels were decreased in all groups at 4 and 12 months post-treatment (*p* < 0.01; Table 2).

### 3.2. TBARS

Compared to baseline, we observed a reduction of TBARS in the whole sample during follow-up (F = 5.23, *p* = 0.007) (Table 2). More specifically, there was a gradual reduction of TBARS values at 4 months (8.27 ± 2.67 μmol/L at baseline vs. 7.94 ± 2.81 μmol/L at 4 months, *p* = 0.022), that was even greater at 12 months of treatment (7.91 ± 2.54 μmol/L, *p* = 0.005). There was a significant interaction of time with treatment regimens (F = 3.10, *p* = 0.045). Patients under the combination GLP1-RA+SGLT2i displayed the most prominent relative reduction of TBARS at four months compared to the other groups (*p* < 0.005 for all comparisons). At 12 months, patients that received GLP1-RA also reduced TBARS significantly but those who received the combination GLP1-RA+SGLT2i demonstrated the most remarkable drop of TBARS compared to those in SGLT2i and insulin group (the relative reduction compared to baseline was −9.83% for GLP1-RA+SGLT2i group; −8.76% for GLP1-RA; 3.22% for SGLT2i and −0.5% for insulin group. The difference between GLP1-RA+SGLT2i and SGLT2i groups was −13.05% (95%CI: −17.25–−8.86%), *p* < 0.001 and the difference between GLP1-RA and SGLT2i groups was −11.98% (95%CI: −15.62–−3.28%), *p* = 0.020. The difference between GLP1-RA+SGLT2i and insulin groups was −10.33% (95%CI: −16.84–−5.23%), *p* = 0.010 and the difference between GLP1-RA and insulin groups was −9.26% (95%CI: −13.48–−2.16%), *p* = 0.029 (Figure 2a).

### 3.3. MDA

We documented decrease of MDA levels in the total sample during follow-up (F = 4.23, *p* = 0.040) (Table 2). These effects were shown at 4 months in the combination group compared with the other three study groups (*p* < 0.05 for all comparisons). The maximal decrease of MDA was documented after 12 months of treatment compared to baseline. Furthermore, there was a significant interaction of treatment with follow-up time (F = 5.54, *p* = 0.005). More specifically, patients treated with either GLP1-RA or the combination GLP1-RA+SGLT2i achieved remarkable reduction of MDA after 12 months of treatment compared to SGLT2i and insulin treatment groups (the relative change compared to baseline was −31.44% for GLP1-RA+SGLT2i group; −30.15% for GLP1-RA; −3.74% for SGLT2i group and 4.72% in insulin group). The difference between GLP1-RA+SGLT2i and SGLT2i groups was −27.70% (95%CI: −36.44–−19.81%), *p* = 0.010 and the difference between GLP1-RA and SGLT2i groups was −26.41% (95%CI: −33.40–−14.26%), *p* = 0.015. The difference between GLP1-RA+SGLT2i and insulin groups was −36.16% (95%CI: −48.73–−28.42%), *p* < 0.001 and the difference between GLP1-RA and insulin groups was −34.87% (95%CI: −43.56–−22.94%), *p* = 0.009 (Figure 2b).

### 3.4. ABTS

All participants exhibited increased levels of ABTS radical scavenging activity during the follow-up, initially at 4 months and even more at 12 months of treatment (F = 4.10, *p* = 0.019) (Table 2). More notably, we noticed a slight elevation of ABTS at 4 months (16.73 ± 4.10 mmol/L at baseline vs. 17.49 ± 4.08 mmol/L at 4 months, *p* = 0.399) but there was a sharp rise at 12 months of treatment (18.08 ± 4.37 mmol/L at 12 months, *p* = 0.019). There was a significant interaction of time with treatment (F = 3.47, *p* = 0.045). Interestingly, among treatment regimens, dual treatment resulted in the higher increase of ABTS at 4 months, in contrary with the rest treatment groups (*p* < 0.05 for all comparisons). After 12 months, patients that received SGLT2i and the combination GLP1-RA+SGLT2i showed significant rise of ABTS levels in contrast with the other 2 groups. The percentage change was 14.13% for GLP1-RA+SGLT2i group and 12.87% for SGLT2i versus −3.44% in GLP1-RA group and 2.44% in the insulin group. The difference between GLP1-RA+SGLT2i and GLP1-RA groups was 17.57% (95%CI: 10.84–23.05%), *p* = 0.010 and the difference between SGLT2i and GLP1-RA groups was 16.31% (95%CI: 7.98–19.64%), *p* = 0.025. The difference between GLP1-RA+SGLT2i and insulin groups was 11.69% (95%CI: 6.57–16.75%), *p* = 0.028 and the difference between SGLT2i and insulin groups was 10.43% (95%CI: 7.92–14.60%), *p* = 0.036 (Figure 2c).

### 3.5. TAC

We recorded a significant interaction between the time of follow-up and treatment regimens (F = 3.25, *p* = 0.028) (Table 2). We observed an increase of TAC in GLP1-RA+SGLT2i group after 12 months of treatment compared to baseline. Conversely TAC levels remained unchanged maintained in the 3 remaining groups at 4 and 12 months [the percentage change at 12 months in GLP1-RA+SGLT2i group was 4.25%; −1.14% in the SGLT2i group, [difference: 5.39% (95%CI: 1.22–11.35%), *p* = 0.028]; −0.45% in the GLP1-RA group, [difference: 4.70% (95%CI: 0.96–8.34%), *p* = 0.040]; −1.08% in the insulin group, [difference: 5.33% (95%CI: 2.78–10.64%) *p* = 0.032].

### 3.6. RP

No statistically significant changes were observed in RP capacity of all participants throughout out the study F = 0.79, *p* = 0.500) (Table 2).

After adjustment for baseline BMI, level of blood lipids, glucose or HbA1 and their respective changes at 4 or 12 months by multivariable analysis, the results remained unchanged for all examined oxidative stress markers (*p* < 0.05 data not shown).

### 3.7. Association of Glycemic Control with Changes of Oxidative Stress Markers

At 4 months, the percentage change of ABTS and TBARS was associated with the changes of HbA1c and fasting glucose, respectively (r = −0.262, *p* = 0.038, and r = 0.231, *p* = 0.049, respectively). At 12 months, the percentage change of ABTS and TBARS correlated with the percentage change of glucose and HbA1C, respectively (r = −0.296, *p* = 0.041 and r = 0.280, *p* = 0.045, respectively) (Table 3). The rest biomarkers did not correlate significantly with the changes of either fasting glucose or Hba1c at 6 and at 12 months of treatment *p* > 0.05, Table 3).

## 4. Discussion

In the present study, we demonstrated that patients treated with GLP1-RA, SGLT2i, and their combination for 12 months achieved remarkable improvement of oxidants and antioxidants compared with those that received insulin, albeit all patients achieved adequate glycemic control, as measured by HbA1C levels. In particular, patients under the combination displayed the greatest improvement of TBARS and MDA, and those under GLP1-RA showed great reduction of TBARS and MDA levels as well. However, the reduction of TBARS and MDA were more prominent and appeared earlier (at 4 months) in GLP1-RA+SGLT2i group in contrast with the GLP1-RA group. On the other hand, patients that received SGLT2i showed remarkable increase of ABTS after 12 months of treatment; also, significant elevation of ABTS displayed those in the combination group as well. Consequently, the favorable effects of combined treatment seem to be synergistic, as GLP1-RA provided reduction of TBARS and MDA and SGLT2i increased ABTS radical scavenging capacity correspondingly. Moreover, only subjects in the dual therapy displayed significant rise of TAC in our study.

T2DM has increased prevalence rate globally and to date has reached epidemic proportions. Cardiovascular complications constitute the major cause of mortality and morbidity in diabetic population and, thus, efforts are being made to identify the underlying mechanisms and, afterwards, organize strategies to counterattack [1,2,3,4,5,39]. Undoubtedly, oxidative stress possesses a key role in the pathogenesis of cardiovascular complications in diabetic patients [6,7,40]. Notably, the term oxidative stress describes a disturbance in balance between the production of reactive oxygen species (ROS) and the antioxidant activity of the body with resultant accumulation of ROS and tissue injury [41]. Although ROS, such as hydrogen peroxide (H_2_O2), have been demonstrated to regulate glucose-stimulated insulin secretion (GSIS) in B-cells, excessive and/or sustained ROS production can affect the integrity and function of cellular macromolecules, such as DNA, protein, or lipids and has been shown to impair the ROS positive signaling function on GSIS [42]. Oxidative stress promotes the development of inflammatory cytokines and thus accelerates inflammation which causes additional tissue damage and in turn induces additional generation of ROS [43]. Of note this process tends to make the atheromatous plaque more vulnerable to rupture [44]. Furthermore, chronic hyperglycemia is thought to enhance the above-mentioned pathways dramatically [45]. The mechanisms predominantly involved in ROS formation in diabetic patients encompass the xanthine oxidase (XO), uncoupled endothelial nitric oxide synthase (eNOS) and NADPH oxidase (NOx) pathways. XO induced ROS cause endothelial dysfunction via hydrogen peroxide production. Also, the eNOS system is elementary for proper endothelial function and in states of excess superoxide generation, NO release by eNOS is depleted and endothelial dysfunction ensues [46,47,48].

SGLT2i are believed to ameliorate oxidative stress burden [11]. Briefly, some of the putative mechanisms include: inactivation of NADPH oxidase, reduction of advanced glycation end products (AGE) owing to glycemic control. More specifically, empagliflozin was proposed to affect NADPH oxidase activity in rat models by interfering with the expression of Nox1 and Nox2 enzymes and suppressing reactive oxygen species production [49]. Likewise, dapagliflozin was found to down-regulate expression of Nox4 and proinflammatory factors in mice [50]. Importantly, decrease of AGE production contributes to attenuation of oxidative stress [51]. Also, in another survey, dapagliflozin was found to inhibit H_2_O2-mediated cell damage and simultaneously decrease the concentration of cytosolic and mitochondrial ROS of human tubular cells in vitro [52]. Furthermore, other studies in rat models treated with ipragliflozin showed drop of TBARS along with decrease of inflammatory cytokines, such as IL-6, IL-1 and TNF-a [53] and drop of plasma and liver TBARS and plasma and liver Protein Carbonyls (PC) as well [54]. Similarly, two other studies conducted in rat models showed that treatment with empagliflozin resulted in reduction of plasma TBARS [55,56]. These findings were not apparent in our study. We observed that TBARS and MDA levels did not drop in SGLT2i group. In a study conducted in rats’ treatment with empagliflozin did not result in comparable reduction of subcutaneous TBARS [57]. One possible explanation is that increased ketogenesis induced by SGLT2i [11] may contribute to lipid peroxidation [58,59]. It should be noted that ketone bodies are considered more effective for myocardial function than free fatty acids, as they provide more ATP per oxygen molecule [12]. Therefore, caution is dictated in interpretation of this biomarker. Additionally, studies investigating the effect of SGLT2i on oxidative stress, documented decrease of urine and serum levels of 8-iso-prostaglandin F2a (8-iso-PGF2a) and 8-Hydroxy-20-deoxyguanosine (8-OHdG) as well, two commonly used biomarkers of oxidative stress burden [60].

Nonetheless, patients treated with SGLT2i in our study displayed great increase of ABTS radical scavenging capacity counteracting the potential oxidative effects of hyperglycemia-induced ketogenesis. Therefore, the net effect of SGLT2 treatment was toward a beneficial equilibrium between oxidant and antioxidant mechanisms as reflected by the unchanged MDA and TBARS levels and the increase of ABTS radical scavenging capacity in this treatment group. Similar changes in ABTS were observed in GLP1-RA+SGLT2i group as well and were attributed to the SGLT2i action. This finding is aligned with the results of another study, which showed the antioxidant properties of SGLT2i empagliflozin by recording significant rise of Glutathione (GSH) content in leukocytes of diabetics after 24 weeks of treatment [61]. Additionally, SGLT2 inhibition by phlorizin attenuated oxidative stress by induction of antioxidants catalase (CAT) and glutathione peroxidase (GPX) activity and reduced nitrogen free radicals in diabetic rats [62]. Also, empagliflozin was proposed to promote translocation of NFE2-related factor 2 (Nrf2) to the nucleus and activation of Nrf2/ARE signaling. Nrf2 constitutes a transcription factor that activates the innate response of human body against ROS [63]. Previous studies reported that administration of SGLT2i led to increased TAC levels. Specifically, in a study in rat models, treatment with dapagliflozin increased levels of TAC, Glutathione Peroxidase (GSH-Px), Superoxide Dismutase (SOD), and at the same time decreased MDA levels. In another study conducted in rats, similar results were observed, as treatment with dapagliflozin resulted in reduction of MDA and elevation of TAC levels [27,28]. This statement is not fully supported by our findings, as MDA and TAC maintained at approximately the same levels in SGLT2i group; albeit we documented increase of TAC in GLP1-RA+SGLT2i group, implicating the robust effects of combined treatment on antioxidative capacity.

As far as the effect of GLP1-RA on markers of oxidative stress is concerned, several mechanisms have been proposed apart from the profound anti-inflammatory and glucose-lowering properties [10,12]. A mechanism that may confer to the amelioration of oxidative stress by GLP1-RA is reduction of TNF-α-induced oxidative stress and inflammation in endothelial cells and this effect is mediated by a calcium- and AMPK-dependent mechanism [64]. We have previously shown that diabetic patients on liraglutide achieved a remarkable reduction of MDA levels after 6 months of treatment, a finding consistent with our current findings and with the results of other studies as well [37,65]. Also, in a study involving rat models, treatment with liraglutide resulted in improvement of oxidative stress by both reducing renal TBARS and downregulating Nox4 and NAD(P)H oxidase activity. Also, they recorded elevation of renal cAMP level and protein kinase A followed liraglutide administration [66]. Furthermore, Chong-Gui Zhu et al. proposed recently that liraglutide attenuates oxidative stress through reduction of TBARS and simultaneously ROS production [67]. Moreover, a study including rats, also proposed that treatment with liraglutide reversed hepatic steatosis and provided pronounced drop of inflammatory factors along with TBARS and ROS [68]. Interestingly, Coagonist of Glucagon-Like Peptide-1 (GLP-1) Receptor (GLP-1R) and Glucagon Receptor (GCGR) in mice suppressed hepatic TBARS, apart from the remarkable reduction of inflammatory cytokines [69]. These suggestions regarding TBARS are in parallel with our findings. Indeed, subjects treated with GLP1-RA displayed a significant decrease of TBARS levels in our study, as well those in dual treatment. Therefore, the favorable effect of combined treatment on oxidative stress was attributed mainly to the GLP1-RA action, although the effects appeared earlier and were more prominent in those under dual therapy.

Of note, in our study we did not record a statistically significant improvement of the antioxidant biomarkers in GLP1-RA, albeit various mechanisms have been described that possibly enhance antioxidative capacity by these agents. Chong-Gui Zhu et al. also suggested that liraglutide up-regulates the NRF2 pathway, which is involved in antioxidant signaling by precipitating the expression of antioxidant-linked genes to repair oxidative injury [67]. Furthermore, others reported that Liraglutide treatment resulted in increase of GSH, although statistically significant changes in the examined redox biomarkers were not obvious [70]. Another presumed mechanism is induction of protein kinase A (PKA) and cAMP response element-binding (CREB) protein with resultant enhancement of antioxidative defenses [71].

The limitations that apply to our study is the relatively small sample size. Additionally, the levels of the examined biomarkers may be influenced by a variety of factors. For instance, TAC levels depend highly on uric acid levels, which may be elevated in diabetes as a byproduct of oxidative-antioxidant reactions [72]. Τhus, further prospective cohorts are warranted to elucidate the salutary effects of combined treatment with GLP1-RA+SGLT2i on oxidative stress biomarkers in diabetic patients.

## 5. Conclusions

In conclusion, a 12-month treatment with GLP-1RA, SGLT-2i, and their combination confer a significant improvement in serum levels of antioxidant biomarkers as well as in products of the detrimental impact of oxidative modifications respectively. These improvements are even more prominent in the combination group, suggesting the synergistic and additive action of these two agents. Therefore, concurrent treatment with GLP1-RA+SGLT2i appears to exceed the separate administration of each agent regarding changes in biomarkers of oxidative stress and antioxidant ability, respectively, and may be incorporated in the management of high cardiovascular risk T2DM patients earlier in the guidelines’ algorithm.

## Figures and Tables

**Figure 1 antioxidants-10-01379-f001:**
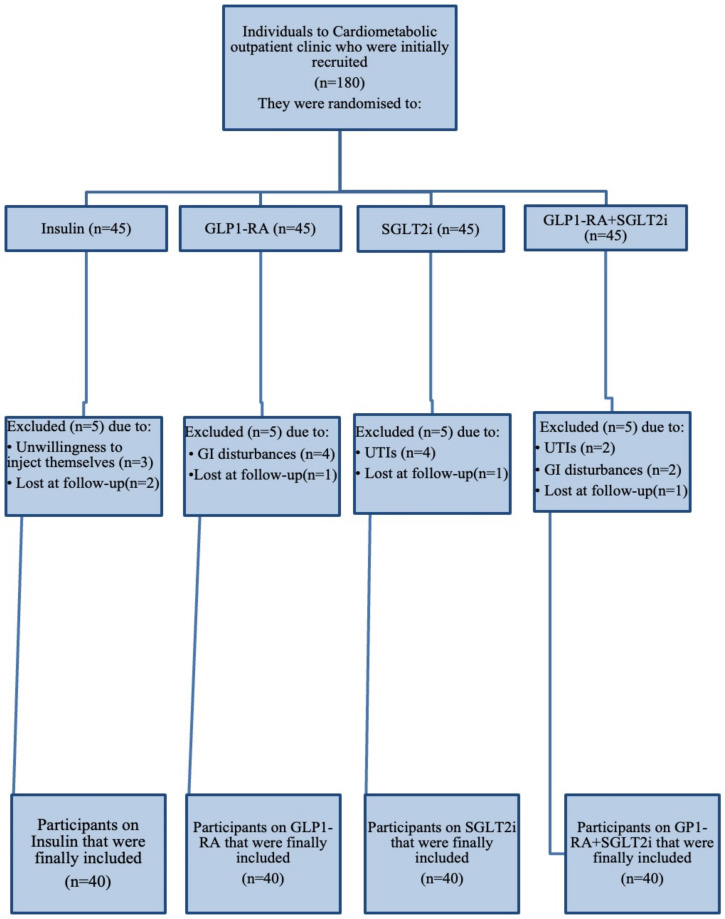
Flowchart of study progress. GLP-1RA: glucagon-like peptide-1 receptor agonists; SGLT-2i: sodium-glucose cotransporter-2 inhibitors.

**Figure 2 antioxidants-10-01379-f002:**
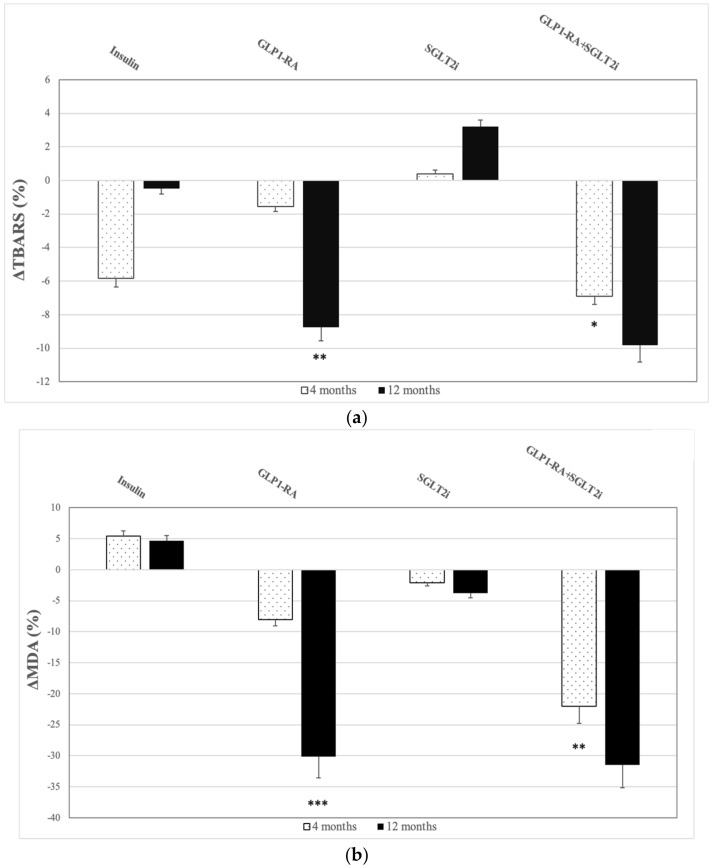
Percentage changes (Δ) from baseline in -(**a**) Thiobarbituric Acid Reactive Substances (TBARS), -(**b**) Malondialdehyde (MDA) and -(**c**) 2,2¢-azino-bis-(3-ethylbenzthiazoline-6-sulphonic acid) radical (ABTS) at 4 and 12 months in the 4 study groups. Data are presented as means ± SD. * *p* < 0.05; ** *p* < 0.01; *** *p* < 0.001 versus baseline.

**Table 1 antioxidants-10-01379-t001:** Data are presented as absolute frequencies (relative frequencies), mean ± SD, or median (first quartile to third quartile). Scale variables were compared with the paired Student *t* test. Binary variables were compared with the χ^2^ test. ACEI indicates angiotensin-converting enzyme inhibitor; ARB, angiotensin receptor blocker; CAD, coronary artery disease; eGFR, estimated glomerular filtration rate; GLP-1RA, glucagon-like peptide-1 receptor agonists; *P*, *P* of model of the ANOVA for comparisons between groups; and SGLT-2i, sodium-glucose cotransporter-2 inhibitors.

	All Patients (*n* = 160)	Insulin (*n* = 40)	GLP-1RA (*n* = 40)	SGLT-2i (*n* = 40)	GLP-1RA+SGLT-2i (*n* = 40)	*p*
**Age**, years	58 ± 10	57 ± 10	57 ± 9	58 ± 10	58 ± 9	0.518
**BMI**, Kg/m^2^	30 ± 3	29.7 ± 3	30 ± 4	29.8 ± 3	30.5 ± 3	0.652
**CAD**, *n* (%)	54 (34)	14 (35)	13 (32.5)	13 (32.5)	14 (35)	0.869
**Creatinine**, mg/dL	1.1 ± 0.3	1.0 ± 0.3	1.1 ± 0.2	1.1 ± 0.2	1.1 ± 0.3	0.833
**Duration of diabetes**, years	6.5 (2–10)	6.7 (1–9)	5.9 (1–8)	6.6 (1–11)	6.8 (2–12)	0.446
**eGFR**, mL/min per 1.73 m^2^	85 ± 10	86 ± 9	85 ± 8	85 ± 10	83 ± 11	0.315
**Fasting plasma glucose**, mg/dL	152 ± 42	158 ± 48	152 ± 45	145 ± 34	151 ± 40	0.474
**Sex (male/female)**, *n* (%)	115/45 (72/28)	28/12 (70/30)	27/13 (67.5/32.5)	30/10 (75/25)	30/10 (75/25)	0.151
**TC**, mg/dL	172 ± 42	178 ± 43	172 ± 38	163 ± 37	175 ± 50	0.237
**LDL-C**, mg/dL	99 ± 26	104 ± 27	101 ± 26	92 ± 21	100 ± 30	0.361
**HDL-C**, mg/dL	42 ± 12	46 ± 12	41 ± 11	39 ± 9	43 ± 14	0.545
**TG**, mg/dL	155 ± 44	143 ± 42	158 ± 41	157 ± 44	161 ± 48	0.198
**Risk factors**, *n* (%)						
**Current smoking**	64 (40)	15 (37.5)	17 (42.5)	16 (40)	16 (40)	0.837
**Dyslipidemia**	160 (100)	40 (100)	40 (100)	40 (100)	40 (100)	1.000
**Family history CAD**	51 (32)	12 (30)	11 (27.5)	15 (37.5)	13 (32.5)	0.392
**Hypertension**	97 (61)	24 (60)	24 (60)	24 (60)	25 (62.5)	0.789
**Cardiovascular medications**, *n* (%)						
**ACEI or ARB**	80 (50)	20 (50)	19 (47.5)	21 (52.5)	20 (50)	0.734
**Aldosterone antagonists**	7 (4)	1 (2.5)	2 (5)	1 (2.5)	3 (7.5)	0.827
**Antiplatelet**	57 (36)	14 (35)	13 (32.5)	15 (37.5)	15 (37.5)	0.898
**Beta blockers**	78 (49)	18 (45)	19 (47.5)	21 (52.5)	20 (50)	0.753
**Calcium channel blocker**	40 (25)	10 (25)	9 (22.5)	10 (25)	11 (27.5)	0.512
**Diuretics**	28 (17.5)	6 (15)	5 (12.5)	8 (20)	9 (22.5)	0.969
**Fibrate**	10 (6)	2 (5)	2 (5)	3 (7.5)	3 (7.5)	0.787
**Statins**	160 (100)	40 (100)	40 (100)	40 (100)	40 (100)	1.000
**Antidiabetic medications**, *n* (%)						
**Metformin**	109 (68)	29 (72.5)	25 (62.5)	27 (67.5)	28 (70)	0.192

**Table 2 antioxidants-10-01379-t002:** Changes in the circulating levels of oxidation stress and antioxidant biomarkers.

Variables	All Participants (*n* = 160)	Insulin (*n* = 40)	GLP1-RA (*n* = 40)	SGLT2i (*n* = 40)	GLP1-RA +SGLT2i (*n* = 40)
**TBARS**, μmol/L					
Baseline	8.27 ± 2.67	8.81 ± 1.22	8.23 ± 2.91	7.76 ± 2.69	8.24 ± 2.38
4 mo	7.94 ± 2.81 ^†^	8.11 ± 1.18	8.10 ± 2.74	7.79 ± 2.58	7.67 ± 1.99 ^†^
Δ%	−3.99	−5.84	−1.57	0.38	−6.91
12 mo	7.91 ± 2.54 ^††^	8.45 ± 1.01	7.06 ± 2.50 ^††,^*	8.01 ± 2.37	7.43 ± 1.94 ^†††,^**
Δ%	−4.35	−0.5	−8.76	3.22	−9.83
**MDA**, nM/L					
Baseline	1.73 ± 0.39	1.48 ± 0.34	1.99 ± 0.46	1.87 ± 0.48	1.59 ± 0.27
4 mo	1.61 ± 0.36 ^†^	1.56 ± 0.41	1.83 ± 0.39	1.83 ± 0.37	1.24 ± 0.32 ^††,^*
Δ%	−6.93	5.40	−8.04	−2.13	−22.01
12 mo	1.45 ± 0.28 ^††^	1.55 ± 0.26	1.39 ± 0.31 ^†††,^**	1.80 ± 0.40	1.09 ± 0.25 ^†††,^***
Δ%	−16.18	4.72	−30.15	−3.74	−31.44
**ABTS**, mmol/L					
Baseline	16.73 ± 4.10	17.18 ± 2.35	17.12 ± 3.02	16.16 ± 4.16	17.49 ± 4.93
4 mo	17.49 ± 4.08	17.90 ± 2.96	17.31 ± 3.59	16.46 ± 4.74	18.27 ± 4.11 ^†^
Δ%	4.72	4.19	3.14	1.85	7.55
12 mo	18.08 ± 4.37 ^†^	17.60 ± 1.99	16.53 ± 3.41	18.24 ± 4.93 ^††^,**	19.06 ± 4.71 ^††,^**
Δ%	10.34	2.44	−3.44	12.87	14.13
**TAC**, mmol/L					
Baseline	0.89 ± 0.14	0.92 ± 0.16	0.84 ± 0.17	0.87 ± 0.12	0.90 ± 0.14
4 mo	0.88 ± 0.16	0.89 ± 0.18	0.82 ± 0.17	0.88 ± 0.16	0.90 ± 0.13
Δ%	−1.12	−3.37	−2.38	1.14	−0.63
12 mo	0.88 ± 0.17	0.91 ± 0.16	0.84 ± 0.16	0.86 ± 0.17	0.94 ± 0.17 ^†,^*
Δ%	−1.12	−1.08	−0.45	−1.14	4.25
**RP**, μmol/mL					
Baseline	1.16 ± 0.14	1.18 ± 0.11	1.18 ± 0.14	1.16 ± 0.13	1.12 ± 0.18
4 mo	1.14 ± 0.13	1.17 ± 0.14	1.17 ± 0.13	1.15 ± 0.12	1.10 ± 0.10
Δ%	−1.72	−1.68	−0.84	−1.72	−1.80
12 mo	1.12 ± 0.17	1.14 ± 0.13	1.14 ± 0.15	1.14 ± 0.14	1.09 ± 0.15
Δ%	−2.97	−3.36	−3.38	−2.69	−2.1
**HbA1c**, %					
Baseline	8.1 ± 1.1	8.2 ± 1.2	8 ± 1.1	7.8 ± 0.9	8.2 ± 1.2
4 mo	6.9 ± 1.1 ^†††^	7 ± 1.1 ^††^	6.7 ± 1 ^†^	7 ± 1 ^†††^	6.7 ± 0.8 ^†††^
Δ%	−17.4	−17.1	−19.4	−11.4	−22.4
12 mo	6.8 ± 1.1 ^†††^	7.1 ± 1.2 ^††^	6.7 ± 0.9 ^†^	7.1 ± 1.1 ^††^	6.4 ± 0.8 ^††^
Δ%	−19.1	−15.5	−19.5	−9.8	−28.1
**Fasting Glucose**, mg/dl					
Baseline	152 ± 42	158 ± 48	152 ± 45	145 ± 34	151 ± 40
4 mo	127 ± 30 ^†††^	134 ± 44 ^†††^	122 ± 27 ^†^	126 ± 20 ^†^	125 ± 29 ^†^
Δ%	−19.73	−17.94	−24.61	−15.10	−20.88
12 mo	120 ± 31 ^††^	121 ± 40 ^†^	118 ± 24 ^†^	124 ± 30 ^†^	116 ± 30 ^†^
Δ%	−26.71	−30.65	−28.80	−16.9	−30.2
**BMI**, kg/m^2^					
Baseline	30 ± 3	29.7 ± 3	30 ± 4	29.8 ± 3	30.5 ± 3
4 mo	28.52 ± 3 ^†††^	28.85 ± 3	28 ± 3 ^†††^	28.82 ± 2 ^†††^	28.54 ± 3 ^†††^
Δ%	−5.31	−3.12	−7.10	−3.51	−7.04
12 mo	28.24 ± 3 ^†††^	29.62 ± 3	27.61 ± 3 ^††^	28.44 ± 2	27.31 ± 4 ^†††^
Δ%	−6.40	−0.32	−8.71	−4.93	−11.71
**TC**, mg/dL					
Baseline	172 ± 42	178 ± 43	172 ± 38	163 ± 37	175 ± 50
4 mo	152 ± 44 ^††^	160 ± 48 ^†^	154 ± 47	140 ± 30 ^†^	154 ± 51 ^†^
Δ%	−13.22	−11.31	−11.71	−16.43	−13.64
12 mo	147 ± 31 ^††^	154 ± 25 ^†^	145 ± 28 ^†^	147 ± 32	144 ± 35 ^††^
Δ%	−17	−15.50	−18.63	−10.92	−21.51
**LDL-C**, mg/dL					
Baseline	99 ± 26	104 ± 27	101 ± 26	92 ± 21	100 ± 30
4 mo	85 ± 31 ^††^	90 ± 35	91 ± 33	78 ± 24 ^†^	79 ± 29 ^†^
Δ%	−16.50	−15.53	−11	−17.91	−26.62
12 mo	80 ± 24 ^††^	82 ± 23	80 ± 21 ^†^	81 ± 25	75 ± 24 ^†^
Δ%	−23.73	−26.80	−26.32	−13.51	−33.44
**HDL-C**, mg/dL					
Baseline	42 ± 12	46 ± 12	41 ± 11	39 ± 9	43 ± 14
4 mo	43 ± 11	46 ± 13	39 ± 7	41 ± 10	45 ± 12
Δ%	2.30	0.42	5.11	4.83	4.41
12 mo	44 ± 10	46 ± 7	42 ± 11	43 ± 8	46 ± 13
Δ%	4.5	0.6	2.4	9.3	6.5
**TG**, mg/dL					
Baseline	155 ± 44	143 ± 42	158 ± 41	157 ± 44	161 ± 48
4 mo	139 ± 37 ^†^	123 ± 28 ^†^	140 ± 38	146 ± 43 ^†^	147 ± 40
Δ%	−11.51	−16.20-	−12.93	−7.52	−10.93
12 mo	122 ± 27 ^††^	118 ± 31	124 ± 22 ^†^	125 ± 29 ^†^	123 ± 24 ^†^
Δ%	−27	−21.12	−27.40	−25.63	−30.92

Data are expressed as mean values ± SD. Δ%: percent changes from baseline. GLP-1RA: glucagon-like peptide-1 receptor agonists; SGLT-2i: sodium-glucose cotransporter-2 inhibitors; TBARS: Thiobarbituric Acid Reactive Substances; MDA: Malondialdehyde; ABTS: 2,2¢-azino-bis-(3-ethylbenzthiazoline-6-sulphonic acid) radical; TAC: Total Antioxidant Capacity; RP: Reducing Power; HbA1c: glycated hemoglobin; BMI: Body Mass Index; TC: Total Cholesterol; LDL-C: Low-Density Lipoprotein Cholesterol; HDL-C: High-Density Lipoprotein Cholesterol; TG: Triglycerides; * *p* < 0.05; ** *p* < 0.01; *** *p* < 0.001 for time × treatment interaction obtained by repeated measures ANOVA. ^†^ *p* < 0.05, ^††^ *p* < 0.01, ^†††^ *p* < 0.001 for comparisons of 4 or 12 months versus baseline by ANOVA using post-hoc analysis with Bonferroni correction.

**Table 3 antioxidants-10-01379-t003:** Correlations of glycemic control with changes of oxidative stress markers. Data are presented as percentages. ΔABTS: the percentage change of 2,2¢-azino-bis-(3-ethylbenzthiazoline-6-sulphonic acid) radical; ΔGlu: the percentage change of fasting plasma glucose; ΔHbA1C: the percentage change of glycated hemoglobin; ΔMDA: the percentage change of Malondialdehyde; ΔRP: the percentage change of Reducing Power; ΔTAC: the percentage change of Total Antioxidant Capacity; ΔTBARS: the percentage change of Thiobarbituric Acid Reactive Substances; N/A: Not Applicable.

	ΔGlu (4 Months)	ΔGlu (12 Months)	ΔHbA1C (4 Months)	ΔHbA1C (12 Months)
ΔABTS (4 months)	r = −0.07 *p* = 0.802	N/A	r = −0.262 *p* = 0.038	N/A
ΔABTS (12 months)	N/A	r = −0.296 *p* = 0.041	N/A	r = −0.138 *p* = 0.265
ΔTBARS (4 months)	r = 0.231 *p* = 0.049	N/A	r = 0.159 *p* = 0.163	N/A
ΔTBARS (12 months)	N/A	r = 0.233 *p* = 0.091	N/A	r = 0.280 *p* = 0.045
ΔTAC (4 months)	r = −0.053 *p* = 0.601	N/A	r = −0.150 *p* = 0.213	N/A
ΔTAC (12 months)	N/A	r = −0.112 *p* = 0.386	N/A	r = −0.104 *p* = 0.401
ΔMDA (4 months)	r = −0.180 *p* = 0.286	N/A	r = −0.164 *p* = 0.195	N/A
ΔMDA (12 months)	N/A	r = −0.181 *p* = 0.295	N/A	r = −0.171 *p* = 0.314
ΔRP (4 months)	r = −0.119 *p* = 0.318	N/A	r = −0.062 *p* = 0.602	N/A
ΔRP (12 months)	N/A	r = −0.136 *p* = 0.248	N/A	r = −0.063 *p* = 0.595

## Data Availability

The datasets generated and/or analysed during the current study are not publicly available due to information that could compromise the privacy of research participants but are available from the corresponding author upon reasonable request. All of other data is contained within the article.

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
