# Peer review of "Effects of a 12-Month Treatment with Glucagon-like Peptide-1 Receptor Agonists, Sodium-Glucose Cotransporter-2 Inhibitors, and Their Combination on Oxidant and Antioxidant Biomarkers in Patients with Type 2 Diabetes"

_antioxidants, 2021, doi:10.3390/antiox10091379_

Round 1

Reviewer 1 Report

This manuscript showed that combination treatments of a GLP-1R agonist and an SGLT2 inhibitor for 12 weeks reduced blood levels of oxidative products and increased antioxidation activities more effectively compared to each single treatment in T2DM subjects. The combination treatments of these anti-diabetic drugs have been shown effectiveness for prevention of cardiovascular and renal impairments in diabetic patients in many previous reports. The results of this study provide a part of the mechanism of action.

Comments

1, The multi-item background data of the subjects shown in Table 1 are very consistent among the four treatment groups. How did authors match it?

2, P4, P24: Authors mentioned in introduction “this effects would be independent of glycemic regulation” and in conclusion “the beneficial effects seems to be independent of glycemic regulation” with results of HbA1c. The author must provide the basis for this explanation and conclusion because it is well known that persistent hyperglycemia results in strong oxidative stress. Reduction of HbA1c levels shown in Table 2 seems to be different among 4 groups, especially the reduction in the combination groups is the largest. Authors should analyze correlation between HbA1c levels and all parameters on oxidation.

3, Figure 2: The difference between a single treatment of GLP-1RA and SGLT2i is interesting. The authors have discussed the effects of GLP-1RA and SGLT2i, citing many previous papers. This reviewer think it would be better to organize the discussion a little shorter and then discuss this point more clearly.

4, P21, the last 2 line: insert “i” after SGLT2. Also, it seems better to replace "SGLT2-induced ketogenesis" at the last 3 line with "hyperglycemia-induced ketogenesis".

Author Response

We would like to thank this reviewer for the constructive comments.

Comments

1, The multi-item background data of the subjects shown in Table 1 are very consistent among the four treatment groups. How did authors match it?

Response: We acknowledge the reviewers point We believe that our study patients’ selection criteria and design of treatment assignment secured a similar background data of patients within each treatment group.

Firstly, as stated in the section of study population we have selected for the purpose of the study diabetic patients with high CV risk ( SCORE ≥5% and <10%), or  very high CV risk (target organ damage or with a major risk factor such as smoking or poorly controlled hyperlipidemia or poorly controlled hypertension -a calculated SCORE ≥10%) indicating a study population with similar atherosclerotic risk factor profile and background medication.

Secondly, we have now further clarified in the revised text that patients were randomly assigned to each treatment regimen by an attending physician (F.K) using a table of random numbers as reproduced from the online randomization software http://www.graphpad.com/quickcalcs/index.cfm as follows (page 5, paragraph 3).

“Patients were randomly assigned to one of the following regimens:  1) basal insulin, with a dose range 10 IU to 50 IU and the dose was titrated according to the running guidelines of that period, (45 of the participants), 2) 1.8 mg of the GLP1-RA liraglutide subcutaneously once daily, which was escalated weekly (45 of the participants), 3) 25 mg of the SGLT2i empagliflozin orally once daily (45 of the participants), and 4) or the combination liraglutide + empagliflozin as a second step treatment to metformin. [29,31-33]. Assignment to a treatment regimen was performed by an attending physician (F.K) using a table of random numbers as reproduced from the online randomization software http://www.graphpad.com/quickcalcs/index.cfm”

Thus, our study selection criteria and design secured a similar background data of patients within each treatment group.

2, P4, P24: Authors mentioned in introduction “these effects would be independent of glycemic regulation” and in conclusion “the beneficial effects seems to be independent of glycemic regulation” with results of HbA1c. The author must provide the basis for this explanation and conclusion because it is well known that persistent hyperglycemia results in strong oxidative stress. Reduction of HbA1c levels shown in Table 2 seems to be different among 4 groups, especially the reduction in the combination groups is the largest. Authors should analyze correlation between HbA1c levels and all parameters on oxidation.

Response: We acknowledge this reviewer’s point as change of TBARS post treatment was associated with the reduction of HbA1, we have now deleted the above statements in introduction and conclusion

The association of between change in HbA1 and change of oxidative markers is now presented in a novel subsection #3.7 (page 23, paragraph 3 of the revised manuscript)and the table3 (pages 24-25) as follows”

3.7 Association of glycemic control with changes of oxidative stress markers

 At 4 months, the percentage change of ABTS and TBARS was associated with the changes of HbA1c and fasting glucose respectively (r=-0.262, p=0.038 and r=0.231, p=0.049 respectively). At 12 months, the percentage change of ABTS and TBARS correlated with the percentage change of glucose and HbA1C respectively  (r=-0.296, p=0.041 and r=0.280, p=0.045 respectively)(Table 3). The rest biomarkers did not correlate significantly with the changes of neither fasting glucose or Hba1c at 6 and at 12 months of treatment p>0.05, Table 3 )

Table 3. Correlations of glycemic control with changes of oxidative stress markers. Data are presented as percentages. ΔABTS: the percentage change of 2,2¢-azino-bis-(3-ethylbenzthiazoline-6-sulphonic acid) radical; ΔGlu: the percentage change of fasting plasma glucose; ΔHbA1C: the percentage change of glycated hemoglobin; ΔMDA: the percentage change of Malondialdehyde; ΔRP: the percentage change of Reducing Power; ΔTAC: the percentage change of Total Antioxidant Capacity; ΔTBARS: the percentage change of Thiobarbituric Acid Reactive Substances;; N/A: Not Applicable

Table 3.

ΔGlu (4 months)

ΔGlu (12 months)

ΔHbA1C (4 months)

ΔHbA1C (12months)

ΔABTS (4 months)

r=-0.07
p=0.802

N/A

r=-0.262
p=0.038

N/A

ΔABTS (12 months)

N/A

r=-0.296
p=0.041

N/A

r=-0.138
p=0.265

ΔTBARS (4 months)

r=0.231
p=0.049

N/A

r=0.159
p=0.163

N/A

ΔTBARS (12 months)

N/A

r=0.233
p=0.091

N/A

r=0.280
p=0.045

ΔTAC (4 months)

r=-0.053
p=0.601

N/A

r=-0.150
p=0.213

N/A

ΔTAC (12 months)

N/A

r=-0.112
p=0.386

N/A

r=-0.104
p=0.401

ΔMDA (4 months)

r=-0.180
p=0.286

N/A

r=-0.164
p=0.195

N/A

ΔMDA (12 months)

N/A

r=-0.181
p=0.295

N/A

r=-0.171
p=0.314

ΔRP (4 months)

r=-0.119
p=0.318

N/A

r=-0.062
p=0.602

N/A

ΔRP (12 months)

N/A

r=-0.136
p=0.248

N/A

r=-0.063
p=0.595

3, Figure 2: The difference between a single treatment of GLP-1RA and SGLT2i is interesting. The authors have discussed the effects of GLP-1RA and SGLT2i, citing many previous papers. This reviewer think it would be better to organize the discussion a little shorter and then discuss this point more clearly.

Response: We acknowledge the reviewer’s point and we have shortened our discussion and focus on the different effects of SGLT2i and GLP1RA on oxidative markers

4, P21, the last 2 line: insert “i” after SGLT2. Also, it seems better to replace "SGLT2-induced ketogenesis" at the last 3 line with "hyperglycemia-induced ketogenesis

Response: Following the reviewer’s suggestion, we have now corrected this phrase to … ketogenesis induced by SGLT2i “ page 27, paragraph 2

Reviewer 2 Report

The manuscript by  Lambadiari et al. investigated the antioxidant capacity and lipid peroxidation levels in diabetic patients under insulin, GLP1-Ra, SGLT2 inhibitor therapies as well as their combination. The results show a superiority of the combined therapy on oxidative stress w/o correlation to HA1C

There is a major controversy whether antioxidant therapy or even focusing on the oxidative environment in diabetes is beneficial.  The discrepancy of the solid in vitro/in vivo data had not correlated with several studies including the HOPE clinical trial, supplementation of vitamin e ext.. Thus, any clinical data regarding the oxidative environment in diabetes and its outcome is of interest.  

  1. The manuscript shows comprehensive bassline biomarkers however, the only clinical data at the end is A1C? why weren’t at least fasting glucose leave as well as a biochemical marker for FFA/ TG/HDL/TC presented. the data can change the conclusions o the Study.
  2. The data suggest that the reduction of antioxidants did not correlate to glycemic control. If so, why (or) is it important to focus on the oxidative state in diabetes?
  3. The discrepancy between the clinical data and in vitro data of antioxidant therapy in the literature should be integrated into the manuscript.
  4. If synergistic is suggested, the authors should calculate CI or change to possible synergistic or additive.

  1. The role of lipid peroxidation and total antioxidant capacity in diabetes had been debated in “several” good manuscripts and reviews. A more extensive introduction on the subject should be added to the introduction.
  2. The source of ROS in diabetes in the introduction lack glucose autoxidation in particular and the MOA of LPO generation should also be elaborated.
  3. The beneficial/signaling aspects of ROS and lipid peroxidation in diabetes should also be debated in the discussion section.
  4. The manuscript had a lot of grammar and typos, for instance:

Christina Chania affiliation is missing

“3.44%),(p<0.05).Only” in the abstract

“Data suggest that diabetics diabetic”

“that was even greater at 12 months of”

key role in the pathogenesis and of cardiovascular complications

The authors are also recommended to rewrite the first paragraph in the introduction. For instance:

Type 2 Diabetes Mellitus is increasing rapidly … increase is strongly

The major underlying mechanism is that diabetes

Author Response

We would also like to thank this reviewer for the critical comments which helped us inprove our manuscript.

  1. The manuscript shows comprehensive bassline biomarkers however, the only clinical data at the end is A1C? why weren’t at least fasting glucose leave as well as a biochemical marker for FFA/ TG/HDL/TC presented. The data can change the conclusions of the Study.

Response: We thank the reviewer for this comment Following the reviewer’s suggestion we have now provided the lipid profile and glucose level at baseline ,4 and 12 months post treatment in the novel table 2 . After adjustment for baseline level of blood lipids, glucose or HbA1 and their respective changes at 4 or 12 months by multivariable analysis, the results remained unchanged for all examined oxidative stress markers. This now stated in the section of results of the revised manuscript as follows”  page 23, paragraph 2.

”After adjustment for baseline BMI, level of blood lipids, glucose or HbA1 and their respective changes at 4 or 12 months by multivariable analysis, the results remained unchanged for all examined oxidative stress markers (p<0.05 data not shown)”

This is now also stated in the section of statistical analysis as follows page 11, paragraph 1

“Baseline BMI, level of blood lipids, glucose or HbA1 and their respective changes at 4 or 12 months were included as covariates in the model.”

Table 2.

Variables

All participants                (n=160)

Insulin

 (n=40)

GLP1-RA

 (n=40)

SGLT2i (n=40)

GLP1-RA +SGLT2i (n=40)

TBARS, μmol/L

Baseline

8.27±2.67

8.81±1.22

8.23±2.91

7.76±2.69

8.24±2.38

4 mo

7.94±2.81

8.11±1.18

8.10±2.74

7.79±2.58

7.67±1.99

Δ%

-3.99

-5.84

-1.57

0.38

-6.91

12 mo

7.91±2.54††

8.45±1.01

7.06±2.50††,*

8.01±2.37

7.43±1.94†††,**

Δ%

-4.35

-0.5

-8.76

3.22

-9.83

MDA, nM/L

Baseline

1.73±0.39

1.48±0.34

1.99±0.46

1.87±0.48

1.59±0.27

4 mo

1.61±0.36

1.56±0.41

1.83±0.39

1.83±0.37

1.24±0.32††,*

Δ%

-6.93

5.40

-8.04

-2.13

-22.01

12 mo

1.45±0.28††

1.55±0.26

1.39±0.31†††,**

1.80±0.40

1.09±0.25†††,***

Δ%

-16.18

4.72

-30.15

-3.74

-31.44

ABTS, mmol /L

Baseline

16.73±4.10

17.18±2.35

17.12±3.02

16.16±4.16

17.49±4.93

4 mo

17.49±4.08

17.90±2.96

17.31±3.59

16.46±4.74

18.27±4.11

Δ%

4.72

4.19

3.14

1.85

7.55

12 mo

18.08±4.37

17.60±1.99

16.53±3.41

18.24±4.93††,**

19.06±4.71††,**

Δ%

10.34

2.44

-3.44

12.87

14.13

TAC, mmol /L

Baseline

0.89±0.14

0.92±0.16

0.84±0.17

0.87±0.12

0.90±0.14

4 mo

0.88±0.16

0.89±0.18

0.82±0.17

0.88±0.16

0.90±0.13

Δ%

-1.12

-3.37

-2.38

1.14

-0.63

12 mo

0.88±0.17

0.91±0.16

0.84±0.16

0.86±0.17

0.94±0.17†,*

Δ%

-1.12

-1.08

-0.45

-1.14

4.25

RP, μmol/ml

Baseline

1.16±0.14

1.18±0.11

1.18±0.14

1.16±0.13

1.12±0.18

4 mo

1.14±0.13

1.17±0.14

1.17±0.13

1.15±0.12

1.10±0.10

Δ%

-1.72

-1.68

-0.84

-1.72

-1.80

12 mo

1.12±0.17

1.14±0.13

1.14±0.15

1.14±0.14

1.09±0.15

Δ%

-2.97

-3.36

-3.38

-2.69

-2.1

HbA1c, %

Baseline

8.1±1.1

8.2±1.2

8±1.1

7.8±0.9

8.2±1.2

4 mo

6.9±1.1†††

7±1.1††

6.7±1

7±1†††

6.7±0.8†††

Δ%

-17.4

-17.1

-19.4

-11.4

-22.4

12 mo

6.8±1.1†††

7.1±1.2††

6.7±0.9

7.1±1.1††

6.4±0.8††

Δ%

-19.1

-15.5

-19.5

-9.8

-28.1

Fasting Glucose, mg/dl

Baseline

152±42

158±48

152±45

145±34

151±40

4 mo

127±30†††

134±44†††

122±27†

126±20

125±29

Δ%

-19.73

-17.94

-24.61

-15.10

-20.88

12 mo

120±31††

121±40

118±24

124±30

116±30

Δ%

-26.71

-30.65

-28.80

-16.9

-30.2

BMI, kg/m2

Baseline

30±3

29.7±3

30±4

29.8±3

30.5±3

4 mo

28.52±3†††

28.85±3

28±3†††

28.82±2†††

28.54±3†††

Δ%

-5.31

-3.12

-7.10

-3.51

-7.04

12 mo

28.24±3†††

29.62±3

27.61±3††

28.44±2

27.31±4†††

Δ%

-6.40

-0.32

-8.71

-4.93

-11.71

TC, mg/dL

Baseline

172±42

178±43

172±38

163±37

175±50

4 mo

152±44††

160±48

154±47

140±30

154±51

Δ%

-13.22

-11.31

-11.71

-16.43

-13.64

12 mo

147±31††

154±25

145±28

147±32

144±35††

Δ%

-17

-15.50

-18.63

-10.92

-21.51

LDL-C, mg/dL

Baseline

99±26

104±27

101±26

92±21

100±30

4 mo

85±31††

90±35

91±33

78±24

79±29

Δ%

-16.50

-15.53

-11

-17.91

-26.62

12 mo

80±24††

82±23

80±21

81±25

75±24

Δ%

-23.73

-26.80

-26.32

-13.51

-33.44

HDL-C, mg/dL

Baseline

42±12

46±12

41±11

39±9

43±14

4 mo

43±11

46±13

39±7

41±10

45±12

Δ%

2.30

0.42

5.11

4.83

4.41

12 mo

44±10

46±7

42±11

43±8

46±13

Δ%

4.5

0.6

2.4

9.3

6.5

TG, mg/dL

Baseline

155±44

143±42

158±41

157±44

161±48

4 mo

139±37

123±28

140±38

146±43

147±40

Δ%

-11.51

-16.20-

-12.93

-7.52

-10.93

12 mo

122±27††

118±31

124±22

125±29

123±24

Δ%

-27

-21.12

-27.40

-25.63

-30.92

  1. The data suggest that the reduction of antioxidants did not correlate to glycemic control. If so, why (or) is it important to focus on the oxidative state in diabetes?

Response: We acknowledge this reviewer’s point. 

The association of between change in HbA1 and change of oxidative markers is now presented in a novel subsection #3.7 (page 23, paragraph 3 of the revised manuscript)and the table3 (pages 24-25) as follows”

3.7 Association of glycemic control with changes of oxidative stress markers

 At 4 months, the percentage change of ABTS and TBARS was associated with the changes of HbA1c and fasting glucose respectively (r=-0.262, p=0.038 and r=0.231, p=0.049 respectively). At 12 months, the percentage change of ABTS and TBARS correlated with the percentage change of glucose and HbA1C respectively  (r=-0.296, p=0.041 and r=0.280, p=0.045 respectively)(Table 3). The rest biomarkers did not correlate significantly with the changes of neither fasting glucose or Hba1c at 6 and at 12 months of treatment p>0.05, Table 3 )

Table 3.

ΔGlu (4 months)

ΔGlu (12 months)

ΔHbA1C (4 months)

ΔHbA1C (12months)

ΔABTS (4 months)

r=-0.07
p=0.802

N/A

r=-0.262
p=0.038

N/A

ΔABTS (12 months)

N/A

r=-0.296
p=0.041

N/A

r=-0.138
p=0.265

ΔTBARS (4 months)

r=0.231
p=0.049

N/A

r=0.159
p=0.163

N/A

ΔTBARS (12 months)

N/A

r=0.233
p=0.091

N/A

r=0.280
p=0.045

ΔTAC (4 months)

r=-0.053
p=0.601

N/A

r=-0.150
p=0.213

N/A

ΔTAC (12 months)

N/A

r=-0.112
p=0.386

N/A

r=-0.104
p=0.401

ΔMDA (4 months)

r=-0.180
p=0.286

N/A

r=-0.164
p=0.195

N/A

ΔMDA (12 months)

N/A

r=-0.181
p=0.295

N/A

r=-0.171
p=0.314

ΔRP (4 months)

r=-0.119
p=0.318

N/A

r=-0.062
p=0.602

N/A

ΔRP (12 months)

N/A

r=-0.136
p=0.248

N/A

r=-0.063
p=0.595

  1. The discrepancy between the clinical data and in vitro data of antioxidant therapy in the literature should be integrated into the manuscript.

Response: Following the reviewers request this discrepancy is now presented in discussion section of the revised manuscript.

  1. If synergistic is suggested, the authors should calculate CI or change to possible synergistic or additive.

Response : Following the reviewer’s suggestion, we have now included in the section of Results  the difference and corresponding 95%confidence intervals of changes in oxidative stress markers between the  combination SGLT2i+GLP1RA  and SGLT2i or GLP1RA regimen

  1. The role of lipid peroxidation and total antioxidant capacity in diabetes had been debated in “several” good manuscripts and reviews. A more extensive introduction on the subject should be added to the introduction.

Response : Following the reviewer’s suggestion, we have now included in the section of Introduction the debate on lipid peroxidation markers ( egMDA) and TAC in diabetes

As follows

“Lipid peroxidation products, are relatively unstable and are the result of a single specific oxidant in contrast to protein carbonyls that represent an irreversible form of

protein modification, are formed early during oxidative stress conditions, are relatively stable and are not a result of one specific oxidant and thus they are considered an accurate marker of overall protein oxidation”

  1. Weber, M.J. Davies, T. Grune, Determination of protein carbonyls in plasma,

cell extracts, tissue homogenates, isolated proteins: focus on sample preparation

and derivatization conditions, Redox Biol. 5 (2015) 367e380, http://

dx.doi.org/10.1016/j.redox.2015.06.005

  1. The source of ROS in diabetes in the introduction lack glucose autoxidation in particular and the MOA of LPO generation should also be elaborated.

Response: Following the reviewer’s suggestion a paragraph on the source of ROS in diabetes, glucose autooxidation and MOA and LPO generation has been included

As follows , page 3 paragraph 1

Six distinct metabolic pathways have be found to contribute to reactive oxygen  (ROS) production in diabetes because of sustained hyperglycemia namely: (1) enolization and α-ketoaldehyde formation, (2) PKC activation, (3) dicarbonyl formation and glycation, (4) sorbitol metabolism, and (5) hexosamine metabolism increasing the oxidative burden and tissue damage including b-pancreatic cells.

A new reference #69 has been included

Fridlyand LE, Philipson LH. Does the glucose-dependent insulin secretion

mechanism itself cause oxidative stress in pancreatic beta-cells? Diabetes. 2004

Aug;53(8):1942-8. doi: 10.2337/diabetes.53.8.1942. PMID: 15277370

  1. The beneficial/signaling aspects of ROS and lipid peroxidation in diabetes should also be debated in the discussion section.

Response: Following the reviewer’s suggestion a paragraph on the beneficial aspects of ROS and lipid peroxidation has been included in Discussionas follows ,page 26, paragraph 2

“Although, ROS, such as H2O2, have been demonstrated  to regulate glucose-stimulated insulin secretion (GSIS) in b-cells, excessive and/or sustained ROS production can affect  the integrity and function of cellular macromolecules, such as DNA, protein, or lipids and has been shown to impair the ROS-positive signaling function on GSIS”

A new reference #71 has been included

Pi J, Bai Y, Zhang Q, Wong V, Floering LM, Daniel K, Reece JM, Deeney JT,

Andersen ME, Corkey BE, Collins S. Reactive oxygen species as a signal in

glucose-stimulated insulin secretion. Diabetes. 2007 Jul;56(7):1783-91. doi:

10.2337/db06-1601. Epub 2007 Mar 30. PMID: 17400930.

  1. The manuscript had a lot of grammar and typos
    Response: Following the reviewer’s suggestion we have corrected all typos 
  2. The authors are also recommended to rewrite the first paragraph in the introduction

    Response: Following the reviewer’s suggestion we have rewritten this paragraph as  follows

    “The incidence of Type 2 Diabetes Mellitus is increasing rapidly worldwide resulting in  increased atherosclerotic cardiovascular complications  and consequently   excess  cardiovascular morbidity and mortality in diabetics compared to the general population”

Reviewer 3 Report

This is a well-written and interesting study that demonstrates the positive effect of GLP-1RA and SGLT2i in the reduction of oxidative stress markers in patients with type 2 diabetes, outstanding the additive effect of GLP-1RA and SGLT2i co-treatment.

Minor comments:

-At least as a supplemental table, the data of other important anthropometric (BMI) and metabolic/clinical parameters (fasting glucose, triglycerides, cholesterol...) at 4 and 12 months should be shown.

-In addition, bivariate correlations (using parametric and/or non-parametric correlations) between the percent change in glycemic measures (HbA1c and fasting glucose) and oxidative stress parameters should be performed and shown to confirm that the reduction in glycemia was not associated with the reduction in oxidative stress parameters.

Author Response

We would  like to extent our thanks to  this reviewer for the critical comments which helped us improve our manuscript.

At least as a supplemental table, the data of other important anthropometric (BMI) and metabolic/clinical parameters (fasting glucose, triglycerides, cholesterol...) at 4 and 12 months should be shown.

Response: Following the reviewer’s suggestion we have now included BMI, glucose and lipid profile in a revised table 2 and the results are also included in text in the section of results

As follows page 14, paragraph 1

“Baseline HbA1C and fating glucose was similar in all participants (p=NS). During follow up all participants achieved significant reduction of HbA1C (F=13.86, p<0.001) and fasting glucose (F=5.1, p=0.013)  and no significant distinctions between treatment groups were recorded (Table 2).  BMI was decreased in the overall population after 4 and 12 months (P<0.001; Table 2). There was a significant interaction between the type of treatment and the change of BMI posttreatment (F=5.939, P for interaction=0.002; Table 2).  Patients treated with combination of GLP-1RA+SGLT-2i presented a greater percentage reduction of BMI compared with GLP-1RA (P=0.042) or SGLT-2i (P=0.009; Table 2) Total cholesterol, low-density lipoprotein cholesterol, and triglycerides blood levels were decreased in all groups at 4 and 12 months post-treatment (P<0.01; Table 2).”

Table 2.

Variables

All participants                (n=160)

Insulin

 (n=40)

GLP1-RA

 (n=40)

SGLT2i (n=40)

GLP1-RA +SGLT2i (n=40)

TBARS, μmol/L

Baseline

8.27±2.67

8.81±1.22

8.23±2.91

7.76±2.69

8.24±2.38

4 mo

7.94±2.81

8.11±1.18

8.10±2.74

7.79±2.58

7.67±1.99

Δ%

-3.99

-5.84

-1.57

0.38

-6.91

12 mo

7.91±2.54††

8.45±1.01

7.06±2.50††,*

8.01±2.37

7.43±1.94†††,**

Δ%

-4.35

-0.5

-8.76

3.22

-9.83

MDA, nM/L

Baseline

1.73±0.39

1.48±0.34

1.99±0.46

1.87±0.48

1.59±0.27

4 mo

1.61±0.36

1.56±0.41

1.83±0.39

1.83±0.37

1.24±0.32††,*

Δ%

-6.93

5.40

-8.04

-2.13

-22.01

12 mo

1.45±0.28††

1.55±0.26

1.39±0.31†††,**

1.80±0.40

1.09±0.25†††,***

Δ%

-16.18

4.72

-30.15

-3.74

-31.44

ABTS, mmol /L

Baseline

16.73±4.10

17.18±2.35

17.12±3.02

16.16±4.16

17.49±4.93

4 mo

17.49±4.08

17.90±2.96

17.31±3.59

16.46±4.74

18.27±4.11

Δ%

4.72

4.19

3.14

1.85

7.55

12 mo

18.08±4.37

17.60±1.99

16.53±3.41

18.24±4.93††,**

19.06±4.71††,**

Δ%

10.34

2.44

-3.44

12.87

14.13

TAC, mmol /L

Baseline

0.89±0.14

0.92±0.16

0.84±0.17

0.87±0.12

0.90±0.14

4 mo

0.88±0.16

0.89±0.18

0.82±0.17

0.88±0.16

0.90±0.13

Δ%

-1.12

-3.37

-2.38

1.14

-0.63

12 mo

0.88±0.17

0.91±0.16

0.84±0.16

0.86±0.17

0.94±0.17†,*

Δ%

-1.12

-1.08

-0.45

-1.14

4.25

RP, μmol/ml

Baseline

1.16±0.14

1.18±0.11

1.18±0.14

1.16±0.13

1.12±0.18

4 mo

1.14±0.13

1.17±0.14

1.17±0.13

1.15±0.12

1.10±0.10

Δ%

-1.72

-1.68

-0.84

-1.72

-1.80

12 mo

1.12±0.17

1.14±0.13

1.14±0.15

1.14±0.14

1.09±0.15

Δ%

-2.97

-3.36

-3.38

-2.69

-2.1

HbA1c, %

Baseline

8.1±1.1

8.2±1.2

8±1.1

7.8±0.9

8.2±1.2

4 mo

6.9±1.1†††

7±1.1††

6.7±1

7±1†††

6.7±0.8†††

Δ%

-17.4

-17.1

-19.4

-11.4

-22.4

12 mo

6.8±1.1†††

7.1±1.2††

6.7±0.9

7.1±1.1††

6.4±0.8††

Δ%

-19.1

-15.5

-19.5

-9.8

-28.1

Fasting Glucose, mg/dl

Baseline

152±42

158±48

152±45

145±34

151±40

4 mo

127±30†††

134±44†††

122±27†

126±20

125±29

Δ%

-19.73

-17.94

-24.61

-15.10

-20.88

12 mo

120±31††

121±40

118±24

124±30

116±30

Δ%

-26.71

-30.65

-28.80

-16.9

-30.2

BMI, kg/m2

Baseline

30±3

29.7±3

30±4

29.8±3

30.5±3

4 mo

28.52±3†††

28.85±3

28±3†††

28.82±2†††

28.54±3†††

Δ%

-5.31

-3.12

-7.10

-3.51

-7.04

12 mo

28.24±3†††

29.62±3

27.61±3††

28.44±2

27.31±4†††

Δ%

-6.40

-0.32

-8.71

-4.93

-11.71

TC, mg/dL

Baseline

172±42

178±43

172±38

163±37

175±50

4 mo

152±44††

160±48

154±47

140±30

154±51

Δ%

-13.22

-11.31

-11.71

-16.43

-13.64

12 mo

147±31††

154±25

145±28

147±32

144±35††

Δ%

-17

-15.50

-18.63

-10.92

-21.51

LDL-C, mg/dL

Baseline

99±26

104±27

101±26

92±21

100±30

4 mo

85±31††

90±35

91±33

78±24

79±29

Δ%

-16.50

-15.53

-11

-17.91

-26.62

12 mo

80±24††

82±23

80±21

81±25

75±24

Δ%

-23.73

-26.80

-26.32

-13.51

-33.44

HDL-C, mg/dL

Baseline

42±12

46±12

41±11

39±9

43±14

4 mo

43±11

46±13

39±7

41±10

45±12

Δ%

2.30

0.42

5.11

4.83

4.41

12 mo

44±10

46±7

42±11

43±8

46±13

Δ%

4.5

0.6

2.4

9.3

6.5

TG, mg/dL

Baseline

155±44

143±42

158±41

157±44

161±48

4 mo

139±37

123±28

140±38

146±43

147±40

Δ%

-11.51

-16.20-

-12.93

-7.52

-10.93

12 mo

122±27††

118±31

124±22

125±29

123±24

Δ%

-27

-21.12

-27.40

-25.63

-30.92

In addition, bivariate correlations (using parametric and/or non-parametric correlations) between the percent change in glycemic measures (HbA1c and fasting glucose) and oxidative stress parameters should be performed and shown to confirm that the reduction in glycemia was not associated with the reduction in oxidative stress parameters.

Response: We acknowledge this reviewer’s point As change of TBARS post treatment was associated with the reduction of HbA1, we have now  deleted the  statements in introduction and conclusion implying that changes of oxidative stress markers was nor associated with reduction of hyperglycemia

The association of between change in HbA1 and change of oxidative markers is now presented in a novel subsection #3.7 (page 23, paragraph 3 of the revised manuscript) as follows”

3.7 Association of glycemic control with changes of oxidative stress markers

 At 4 months, the percentage change of ABTS and TBARS was associated with the changes of HbA1c and fasting glucose respectively (r=-0.262, p=0.038 and r=0.231, p=0.049 respectively). At 12 months, the percentage change of ABTS and TBARS correlated with the percentage change of glucose and HbA1C respectively  (r=-0.296, p=0.041 and r=0.280, p=0.045 respectively)(Table 3). The rest biomarkers did not correlate significantly with the changes of neither fasting glucose or Hba1c at 6 and at 12 months of treatment p>0.05, Table 3 ) 

Table 3.

ΔGlu (4 months)

ΔGlu (12 months)

ΔHbA1C (4 months)

ΔHbA1C (12months)

ΔABTS (4 months)

r=-0.07
p=0.802

N/A

r=-0.262
p=0.038

N/A

ΔABTS (12 months)

N/A

r=-0.296
p=0.041

N/A

r=-0.138
p=0.265

ΔTBARS (4 months)

r=0.231
p=0.049

N/A

r=0.159
p=0.163

N/A

ΔTBARS (12 months)

N/A

r=0.233
p=0.091

N/A

r=0.280
p=0.045

ΔTAC (4 months)

r=-0.053
p=0.601

N/A

r=-0.150
p=0.213

N/A

ΔTAC (12 months)

N/A

r=-0.112
p=0.386

N/A

r=-0.104
p=0.401

ΔMDA (4 months)

r=-0.180
p=0.286

N/A

r=-0.164
p=0.195

N/A

ΔMDA (12 months)

N/A

r=-0.181
p=0.295

N/A

r=-0.171
p=0.314

ΔRP (4 months)

r=-0.119
p=0.318

N/A

r=-0.062
p=0.602

N/A

ΔRP (12 months)

N/A

r=-0.136
p=0.248

N/A

r=-0.063
p=0.595

Round 2

Reviewer 1 Report

Authors responded sufficiently for comments, and the manuscript has been satisfactory revised. 

Reviewer 2 Report

The authors had made considerable modifications and the manuscript can be published in its current form